# Identification of Neoadjuvant Chemotherapy Response in Muscle-Invasive Bladder Cancer by Fourier-Transform Infrared Micro-Imaging

**DOI:** 10.3390/cancers14010021

**Published:** 2021-12-21

**Authors:** Camille Mazza, Vincent Gaydou, Jean-Christophe Eymard, Philippe Birembaut, Valérie Untereiner, Jean-François Côté, Isabelle Brocheriou, David Coeffic, Philippe Villena, Stéphane Larré, Vincent Vuiblet, Olivier Piot

**Affiliations:** 1Jean Godinot Institute, 51100 Reims, France; camillemazza@hotmail.fr (C.M.); jc.eymard@reims.unicancer.fr (J.-C.E.); 2BioSpecT (Translational BioSpectroscopy) EA 7506, SFR Santé, Université de Reims Champagne-Ardenne, 51100 Reims, France; gaydouv@hotmail.com (V.G.); slarre@chu-reims.fr (S.L.); 3Department of Biopathology, University Hospital of Reims, 51100 Reims, France; pbirembaut@chu-reims.fr; 4Cellular and Tissular Imaging Platform (PICT), Université de Reims Champagne-Ardenne, 51100 Reims, France; valerie.untereiner@univ-reims.fr; 5Department of Biopathology, Hôpital de la Pitié-Salpêtrière, APHP, 51100 Paris, France; jean-francois.cote@aphp.fr (J.-F.C.); isabelle-brocheriou@aphp.fr (I.B.); 6Polyclinique Courlancy, 51100 Reims, France; dcoeffic@iccreims.fr (D.C.); pvillena@iccreims.fr (P.V.); 7Department of Urology, University Hospital of Reims, 51100 Reims, France

**Keywords:** muscle-invasive bladder cancer, neoadjuvant chemotherapy, mid-infrared imaging, chemometric algorithms, predictive response to treatment

## Abstract

**Simple Summary:**

Assessing the tumor response to chemotherapy is a paramount predictive step to improve patient care. Infrared spectroscopy probes the chemical composition of samples, and in combination with statistical multivariate processing, presents the capacity to highlight subtle molecular alterations associated with malignancy characteristics. Microscopic infrared imaging of tissue samples reveals spectral heterogeneity within histological structures, providing a new approach to characterize tumoral heterogeneity. We have taken advantage of the analytical capabilities of mid-infrared spectral imaging to implement a classification model to predict the response of a tumor to chemotherapy. Our development was demonstrated in muscle-invasive bladder cancer (MIBC) by comparing samples from responders and non-responders to neoadjuvant chemotherapy.

**Abstract:**

Background: Neoadjuvant chemotherapy (NAC) improves survival in responder patients. However, for non-responders, the treatment represents an ineffective exposure to chemotherapy and its potential adverse events. Predicting the response to treatment is a major issue in the therapeutic management of patients, particularly for patients with muscle-invasive bladder cancer. Methods: Tissue samples of trans-urethral resection of bladder tumor collected at the diagnosis time, were analyzed by mid-infrared imaging. A sequence of spectral data processing was implemented for automatic recognition of informative pixels and scoring each pixel according to a continuous scale (from 0 to 10) associated with the response to NAC. The ground truth status of the responder or non-responder was based on histopathological examination of the samples. Results: Although the TMA spots of tumors appeared histologically homogeneous, the infrared approach highlighted spectral heterogeneity. Both the quantification of this heterogeneity and the scoring of the NAC response at the pixel level were used to construct sensitivity and specificity maps from which decision criteria can be extracted to classify cancerous samples. Conclusions: This proof-of-concept appears as the first to evaluate the potential of the mid-infrared approach for the prediction of response to neoadjuvant chemotherapy in MIBC tissues.

## 1. Introduction

Muscle-invasive bladder cancer (MIBC) is an aggressive disease with a specific survival of 60% at 5 years. It represents 15 to 25% of all bladder carcinomas, which are the 7th incident cancer in the world, responsible for 3% of all cancer deaths [1].

The standard of care of localized MIBC is radical cystectomy [2]. However, 50% of patients treated with surgery alone relapse with metastases within 5 years of follow-up. Adding neoadjuvant chemotherapy (NAC) with cisplatin before surgery has shown an increase in overall survival (OS) and disease-free survival of 5% and 9% respectively [3]. Indeed, NAC permits to reach a survival rate of 85% at 5 years [4,5]. Nonetheless, cisplatin-based-NAC presents a weak average response rate of about 29%. The limited response is likely to be due to tumor heterogeneity. Consequently, more than 70% of patients treated by cisplatin-based-NAC are exposed to chemotherapy and its potential adverse events without any efficacy [6,7].

Histological response to NAC is defined by the absence of invasive tumor after NAC and radical cystectomy. To better understand the relationship between tumor heterogeneity and the sensitivity to chemotherapy, the development of predictive tools to guide clinicians is required [8,9]. Recently, better characterization of tumor heterogeneity has been possible thanks to molecular biology [10]. Several independent research groups have identified MIBC subtypes based on similar biological characteristics, and subgroups associated with primary chemoresistance [11,12,13]. Despite these molecular biology tools, to date, none of these predictive biomarkers and classifications has been sufficiently accurate and validated in a prospective randomized trial. 

Thus, distinguishing responders (R) from non-responders (NR) MIBC samples to NAC represents a real challenge that would allow guiding the therapeutic strategy in localized MIBC. In this context, vibrational FTIR (Fourier Transform InfraRed) spectroscopy appears as a candidate alternative technique to molecular biology. Based on non-destructive interaction between light and matter, FTIR spectroscopy permits to probe the mid-infrared absorption of chemical bonds which provides valuable information on the intrinsic molecular composition of a sample, in a label-free approach. This method can be combined with an imaging setup, thus offering the possibility to lead histopathological characterization of cancer tissues [14,15,16]. Thus, the analytical capacities of the imaging spectral approach have been used for diagnostic purposes by identifying tumor sites but also tumor-associated features [17,18,19,20,21]. In addition, FTIR spectroscopy was used to investigate mechanisms associated with drug response in cell models, such as ovarian and prostate cancer cells [22,23]. Despite the wide scientific literature demonstrating the potential of IR spectroscopy for applications in cancerology, to our knowledge, no study has been referenced on the contribution of this biophotonic technique to evaluate the response to chemotherapy using tissue samples of patients. Thus, this translational study aims to assess the potential of IR micro-imaging to discriminate responders from non-responders to MIBC patients to NAC.

## 2. Materials and Methods

In this retrospective multicentric study, we included all patients treated for localized MIBC with chemotherapy followed by radical cystectomy between January 2012 and October 2016, from Jean Godinot Institute (Reims), Reims University Hospital, Courlancy Institute (Reims) and La Pitié Salpêtrière University Hospital (Paris). The samples were obtained from the tumor banks of these institutes in agreement with the head of the pathology departments concerned. 

Demographical, clinical and histological data were collected for each patient. In this study, “responders” patients were categorized as responders in regards to histological characteristics associated with NAC response, which is defined by the absence of invasive tumor after both NAC and radical cystectomy within the piece of cystectomy. Complete pathological response (pT0) is defined not only by the absence of invasive tumor but also the absence of non-invasive tumor, after NAC and radical cystectomy. The samples were anonymized before their infrared analysis.

### 2.1. Sample Preparation by Tissue Microarray (TMA)

Tissue samples obtained from fixed and paraffin-embedded trans-urethral resection of bladder tumor (TURBT) at time of diagnosis (before starting chemotherapy treatment) were used for the construction of tissue microarray (TMA) using a 1 mm-diameter spot. From the TMA, 10 μm-thin sections were cut and deposited on CaF_2_ supports appropriated for mid-infrared transmission analysis. 

### 2.2. Infrared Microimaging

The IR acquisitions were performed with a Spectrum Spotlight 300 FT-IR Imaging System (Perkin-Elmer, Villebon sur Yvette, France), using a matrix of 16 pixels. Each pixel can be considered as an individual IR detector, analyzing a surface on the sample equivalent to 6.25 µm × 6.25 µm. Spectra were collected from 400 to 4000 cm^−1^ wavelength range, with a 4 cm^−1^ spectral resolution and eight accumulations per measurement. The background was recorded on a sample-free CaF_2_ area. Paraffin signal was also collected with the same parameters on an area surrounding the TMA spots without tissue. 

### 2.3. Chemometric Processing of IR Data

In view to establishing a spectroscopic scale based on IR signatures of the MIBC according to the response to NAC, we adapted chemometric algorithms previously developed for building an invasiveness scale in lung squamous cell carcinoma [24]. In addition to the signal of tissue, the spectral data present also undesirable contributions, such as paraffin interference since no chemical dewaxing was performed before IR acquisitions. To correct such parasitic interferences and normalize the data, EMSC (Extended Multiplicative Signal Correction) preprocessing was first applied, this pre-processing included in particular a mathematical correction of the spectral interferences originated from paraffin embedding [15,17,25]. From this step, the spectral range was reduced to 800–1800 cm^−1^ as it corresponds to the informative fingerprint region with limited spectral interferences. The effect of the EMSC pre-processing can be seen in Appendix A representing the spectral variability of the whole dataset normalized according to the target spectrum (mean spectrum of the dataset). Secondly, data were processed by Kmeans clustering and PLS-DA (Partial Least Square-Discriminant Analysis) to develop an automatic selection of relevant pixels by excluding outliers pixels, i.e., selection of pixels for the subsequent PLS (Partial Least Square) modeling. Finally, a PLS algorithm was implemented to construct a quantitative model permitting to score the spectral data on a numerical scale associated with the response to NAC. This model was developed and optimized by considering image-based cross-validation, in which all pixels/spectra of one image constituted a unique independent sample. 

#### 2.3.1. Individual Kmeans Clustering for Associating IR Spectral Signatures with Tissue Structures

The Kmeans clustering method is an unsupervised analysis method based on the use of an iterative algorithm. This analysis allows to group all the spectra of an image in *K* clusters (number of clusters usually determined by the user) according to a minimum Euclidean distance criterion between each spectrum and the closest cluster center (spectra centroid). This analysis allows reconstructing color-coded images where each cluster is associated with a color. By confronting Kmeans images with standard histology analysis, it is possible to associate clusters with specific tissue structures. So, the spectral images of the calibration set were confronted with HES staining performed on the adjacent slides. In this study, we used a more accomplished version of Kmeans developed in our laboratory [26]. This algorithm presents the advantage to determine automatically the optimal number *K* of clusters. It was applied individually to each image of the TMA. 

#### 2.3.2. Partial Least Square-Discriminant Analysis (PLS-DA) for Automatic Selection of Image IR Pixels

PLS-DA algorithm was employed to implement qualitative classification models. In this study, it can be considered as a quality test to avoid outliers pixels likely to parasite the subsequent PLS analysis (see next paragraph), PLS-DA was, therefore, used for automatic selection of pixels to be scored according to the NAC response scale. For every qualitative variable, a binary code (0 or 1) is associated. A multivariate regression PLS is then realized between the spectral data matrix and the binary matrix. Then, the PLS-DA model predicts multivariate values linked to spectra present in the dataset. The obtained multivariate values are then correlated to corresponding qualitative groups. 

#### 2.3.3. Partial Least Square (PLS) Modeling for Scoring the Response to NAC Based on the IR Signatures of the Tissue Specimens

The PLS approach was employed to associate to each pixel of an image a score associated with the responder (R)/non-responder (NR) status. Here, a scale from 0 to 10 was used to take into account the spectral heterogeneity of the samples, with the reference values for NR and R fixed to 1 and 9 respectively. The PLS algorithms are based on a multivariate regression principle and allow to maximize the covariance between 2 matrixes through multidimensional and orthonormal regression vectors spaces [27]. 

In our study, the calibration and optimization of the PLS model were based on a leave-one-out cross-validation method carried out at the level of the spectral images (rather than at the pixel level to avoid overfitting). The principle of cross-validation is based on the prediction of one sample (one spectral image in our case) beforehand removed from the calibration set, the model is built with the remaining samples. This process is repeated for all the samples of the calibration set. An averaged root mean square error (RMSE) of prediction is then obtained. Based on this average value, it is possible to determine the optimal dimension of the vectorial space, i.e., the number of regression vectors to minimize the RMSE. RMSE was calculated as follows:$$\mathrm{RMSE}=\sqrt{\frac{\sum_{i=1}^{n}{\left(y_{i}-y_{Hi}\right)}^{2}}{n-1}}$$
with *y_i_* being the reference value for the *i*th spectrum, *y_Hi_* being the predicted value of the *i*th spectrum, and *n* being the number of spectra.

#### 2.3.4. External Validation Set

A set of external validation samples was used to test the IR model on independent data not previously used for the construction of the model and whose response/non-response status was unknown at the time of prediction. In our approach, for each TMA spot, a new image was obtained in which the score of each pixel of the image was predicted between 0 and 10 according to the R/NR scale.

#### 2.3.5. Sensitivity and Specificity of the IR Approach According to the Percentage of Pixels and the Responder/Non-Responder Score

From the PLS images of the test set, the outcome of the IR-based predictive approach in terms of sensitivity or specificity depended, therefore, on two criteria: (1) the R/NR score corresponding, for each pixel of an image, to the PLS score on the Responder/Non-Responder scale and (2) the percentage of pixels in an image whose score was below a given R/NR threshold score. Sensitivity corresponds to the ratio of the IR-predicted responders over the total number of responders based on the histological examination of the samples. Specificity is the ratio of IR-predicted non-responders over the total number of non-responders. 

All the computing steps were processed on Matlab R2013a (32 bit) (Mathwork, Nantick, MA, USA), the PLS algorithm originates from “saisir” toolbox developed by Bertrand and Cordella, INRA, France.

## 3. Results

### 3.1. Patients Characteristics

A total of 56 patients were first screened for this proof of concept study. Thirteen patients were excluded due to lack of quality or quantity of their tumor tissue from the TURBT, so 43 patients were finally included in the analysis. Patients, tumor, and treatment characteristics are presented in Table 1 and Table 2. The mean age at diagnosis was 66 years, and 77% of patients were male. Smoking concerned 85% of the patients’ cohort. All patients had ≥ pT2 disease before NAC. The mean number of chemotherapy cycles was 4 and the mean period between last chemotherapy and curative surgery was 40 days. Discontinuation of treatment is not reported due to missing data. The mean follow-up was 22 months (6–54). Among 19 (44%) responder patients, 14 (74%) had complete pathological response (pT0) and 26% had incomplete response (≤pT2 but >pT0).

### 3.2. IR Analysis of the Transurethral Resection of Bladder Tumor Samples and Constitution of Calibration and External Validation Sets

FTIR imaging was carried out on all samples, one by one, and provided 2,789,796 spectra containing potentially valuable information on the intrinsic molecular composition of TURBT tissues. Data were first pre-processed by using EMSC to neutralize undesirable interferences and normalize data according to a target spectrum (mean spectrum of the dataset). Then, to build the predictive PLS-DA (Partial Least Square—Discriminant Analysis) and PLS (Partial Least Square) models and to estimate their performances, the samples were divided into two data sets: calibration set and external validation set. For mimicking a clinical application of our approach, data of each patient (i.e., all images corresponding to the same patient) were allocated either to calibration set or to test set. Thus, 79 images corresponding to 43 patients were separated into two data sets: 42 images in calibration (22 patients) and 37 images in external validation (21 patients). Responder and non-responder patients were divided equitably among the calibration and test sets to maintain homogeneity of the groups.

### 3.3. Recognition of Tissue Structures Using Individual KMeans Clustering and PLS-DA of Spectral Images

On each image of the calibration samples, spectral processing by individual Kmeans clustering was applied for recognition of the tissue architecture. The selection of clusters (or groups of pixels/spectra) of interest is a step necessary for the subsequent construction of the PLS model. This Kmeans treatment permits the identification of different histological structures. Based on their spectral signatures, we report in Figure 1A,B (R sample) and Figure 1E,F (NR sample) that relevant histological structures were identified, such as an invasive tumor, connective tissue, or muscle. The color-coded Kmeans images were constructed using an advanced version of the clustering that determines automatically the number of the clusters [26]. This algorithm permits a fine description of the tissue architecture. However, here the objective is to associate characteristic spectral signatures to the main tissue structures constituting the TURBT samples. So, from the color-coded Kmeans, only the majority clusters were considered for further analysis. For each image, keeping the 6 majority clusters appeared appropriate to retain the large majority of the tissue structures of interest. Note also that minor clusters can be considered as outliers since their constituting pixels are very weakly representative (representing a proportion of pixels less than 1%) and very often localized at the tissue edges. Then, from the Kmeans selected clusters, a PLS-DA model was constructed for the automatic identification of pixels of interest in the predictive approach implemented here. The efficiency of this model can be first evaluated on the same samples of the calibration set, as shown in Figure 1C,G. Pixels colored in black corresponded to non-selected pixels. Orange pixels were identified using PLS-DA for subsequent scoring on the R/NR scale. The strong correspondence between these orange pixels and the majority of Kmeans clusters can be noted for these calibration samples, as it can be noticed by visual comparison of Figure 2B,C (R sample) or Figure 2F,G (NR Sample). 

### 3.4. PLS Scoring of the R/NR Scale

PLS model was built on previously selected pixels of samples from the calibration set. PLS is a process offering the possibility to consider continuous and quantitative scores. Thus, PLS was run with the TMA spots allocated to calibration as reference inputs, by considering a scale from 0 to 10, with the scores of 1 and 9 for NR and R, respectively. Results for the two representative samples (NR and R) are depicted in Figure 1D,H. Globally for all samples of the calibration set, the RMSE achieved a value of 3.68 in cross-validation that corresponded to the best results obtained with a number of latent variables equal to 12. This dimension of the vectorial space, corresponding to the first minimum of the RMSE, is considered appropriate to avoid under and over-fitting for the PLS model. More interestingly, in a further step to demonstrate the validity of our approach, the PLS model was applied to independent TMA spots samples of the external validation set as shown in Figure 2B,E. Black pixels corresponded to non-selected pixels by PLS-DA; this automated process of pixels rejection being considered as a quality test on these samples of the test set. The remaining pixels were colored according to the R/NR scale. 

In addition, PLS processing gives access for each TMA spot to a histogram that represents the distribution of the pixels as a function of their score. Such representation allows evaluating quantitatively the spectral heterogeneity of the tissues (Figure 2C,F). Histograms from the external validation set were then used to build maps of sensitivity and specificity according to the two parameters, R/NR score and percentage of pixels in the image whose score is below a given R/NR threshold score. Indeed, for each couple of these parameters values, sensitivity and specificity were calculated. All these data made it possible to draw 2D maps (Figure 3A,B) displaying the values of sensitivity and specificity obtained as a function of the two parameters.

### 3.5. Sensitivity and Specificity Maps

From these maps (Figure 3), it is possible to maximize either sensitivity or specificity by selecting specific values of both R/NR PLS score and percentage of pixels. For example, from Figure 3B, we see that specificity to identify non-responders patients can be maximized when 10% of the pixels present an R/NR score below 4 or 80% of the pixels with a score below 1 as indicated by the black crosses positioned on the map. Interestingly, Figure 3 showed that the sensitivity and specificity values were represented as sigmoidal curves. This might reflect that the most important variable in our model is the R/NR PLS scoring, the percentage of pixels having less influence.

### 3.6. Spectral Features Underlying the PLS R/NR Scale

Visualization of the latent variables of the PLS model gives access to infrared wavenumbers retained to construct the R/NR spectral scale. Figure 4 displays the mean of the twelve latent variables of the PLS model together with the minimum to maximum space of variability of these latent variables represented by the shaded area. The spectral regions containing the most informative wavenumbers can be highlighted: especially 885–910 cm^−1^ assigned to vibrations of phosphorylated proteins, 1125 cm^−1^ vibration of carbohydrates, and Amide bands corresponding to vibrations of peptide bonds and informative of the secondary structure of the proteins (Amide III between 1200–1350 cm^−1^, Amide II between 1500–1600 cm^−1^ and Amide I between 1600–1700 cm^−1^). The vibration associated with the wavenumber at 1740 cm^−1^ is likely to be assigned to the ester bond in lipids. 

## 4. Discussion

The use of FTIR micro-imaging on MIBC tissue samples was experimented to provide a prognostic spectral marker of NAC response. A predictive score of treatment response would allow a better therapeutic strategy especially by avoiding treating non-responder patients with potentially toxic chemotherapy. Our results showed some significant potential of this vibrational spectroscopic tissue imaging technique to classify R and NR patients and to predict NAC non-responder patients.

So far, the interest in infrared vibrational spectroscopy has been widely demonstrated in diagnostic purposes in various medical disciplines [28,29,30,31,32] and especially in the field of oncology [33,34]. However, to our knowledge, the predictive potential of FTIR micro-imaging on tissue samples has not been published in the literature. To conduct our study, we based the methodology on the development of our group about the implementation of an aggressiveness score for squamous cell lung carcinoma [24].

Our work is reinforced by the use of TMA from TURBT at diagnosis time to obtain histologically-homogeneous tumor samples. Indeed, areas of interest were defined by an experienced genitourinary pathologist. Despite the histological homogeneity of TMA spots, we reported a spectral heterogeneity that could be a quantitative indicator of the sensitivity to chemotherapy. We highlighted the spectral heterogeneity by using a Partial Least Square modeling that provides sensitivity to chemotherapy score for each pixel of the spectral image with a micrometric dimension. Such important valuable information is unreachable by classical histology examination. This is an important issue to bring a better understanding of intratumoral heterogeneity. Indeed, biological intratumoral heterogeneity has been shown to lead to treatment resistance [35,36,37]. Interestingly, the spectral heterogeneity that reflects biochemical heterogeneity, seems to be associated with different NAC responses. Indeed, it seems clinically relevant that only a small percentage of tumors resistant to treatment can lead to patient relapse.

Furthermore, we developed a model built from advanced chemometric algorithms using robust validation. Regarding our population, data for calibration and external validation sets were rigorously selected. Indeed, each patient was either allocated to the calibration set or to the external validation set. The populations of the different datasets were also homogeneous in regards to demographic characteristics, such as sex, age, and smoking habit. The implemented predictive model allows the adjustment of two main variables, such as R/NR scoring and percentage of pixels, leading to the possibility of maximizing sensitivity or specificity according to the clinical objective. Indeed, criteria can be adjusted to favor a way of establishing a screening test. In the current clinical therapeutic strategy, patients are defined as responders or non-responders to chemotherapy according to histological examination. However, the on/off response criteria used in daily practice is not the reflection of the real tissue status, because there is a continuum between response and no response. The tumor is structured with cells having different degrees of response to chemotherapy. In our case, the clinical challenge is to predict non-responder patients, without misidentifying responders that could benefit from chemotherapy. Thus, we developed a model promoting specificity to correctly identify non-responders. A lack of specificity means missing NR patients and wrongly giving them chemotherapy: this is what is done nowadays according to the recommendations and without any predictive test available.

However, our methodological demonstration does not allow immediate implementation of FTIR micro-imaging in clinical practice. Several elements support this opinion. The method used to construct the predictive model requires gold standard references, which in our study corresponded to histological data. Our primary endpoint was a histologic response to chemotherapy defined as the absence of residual invasive tumor (≤pT2) [1,38]. This criterion is usually used in studies evaluating predictive response markers but it can still be discussed [39]. Disease staging obtained at the time of TURBT is pT2 at least, defining invasive urothelial cancers that require NAC followed by radical cystectomy, but it doesn’t allow to evaluate the involvement of perivesical fat (T3) or neighbors organs (T4), leading to possible discrepancies between clinical and pathological stages [40,41]. A precision among non-responders to chemotherapy was recently introduced by a study showing that non-responders could be subdivided into a group of “progressors” and a group corresponding to “stable disease”. The survival was better in the responder group compared to non-responders, but patients in the “stable disease” group presented also better survival than “progressor” patients, who would then constitute the real subgroup of non-responders to chemotherapy for whom a therapeutic alternative is essential [42]. Future investigations will have to take into account this response variability more accurately.

The limited number of patients can be a weakness in our study because it did not permit to separate the non-responder population into subgroups depending on the progression or stability of the disease under chemotherapy. Indeed, despite recommendations, only a weak proportion of patients with localized MIBC received NAC, which explains the limited cohort of this study despite the great number of screened patients [43].

We chose to use TMA tissue samples to focus on the cellular component of the tumor. However, tumor cells are probably not the only determinants of treatment response, which is probably conditioned by the peritumoral environment (stroma) [9,44]. Indeed, it has been shown over the past years that the tumor microenvironment plays an important role in tumor initiation, progression, and metastases, as well as responses to treatment [45,46]. The heterogeneity of the tumor microenvironment is also an issue to better understanding mechanisms of resistance [47]. Thus, despite the undeniable advantage linked to the selection of zones of interest, TMA could hinder the representativeness of the peritumoral heterogeneity that could be taken into account to refine the IR predictive model.

## 5. Conclusions

To our knowledge, this work is the first to evaluate the combination of FTIR imaging of tissue samples with chemometric methods for the discrimination of responders and non-responders patients to chemotherapy in MIBC. The development of this technique, alone or in combination with molecular biology, could guide the therapeutic strategy by targeting the indication for neoadjuvant chemotherapy to responders only.

## Figures and Tables

**Figure 1 cancers-14-00021-f001:**
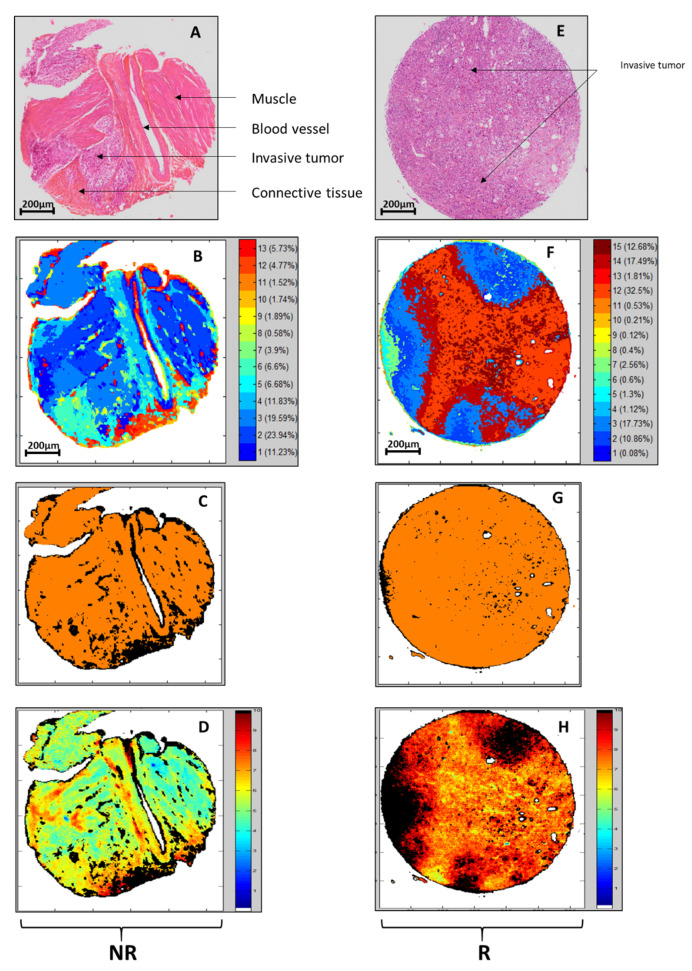
Histological images (**A**,**E**) and chemometric steps; including individual Kmeans clustering (**B**,**F**) for tissue structures recovering; PLS-DA (**C**,**G**) for automatic selection of pixels of interest and R/NR PLS scoring (**D**,**H**); for two representative calibration samples corresponding to NR and R patients. PLS was run with the infrared images of TMA spots allocated to calibration set as reference inputs; by considering a scale from 0 to 1 with scores of 1 (blue) and 9 (red) for NR and R respectively.

**Figure 2 cancers-14-00021-f002:**
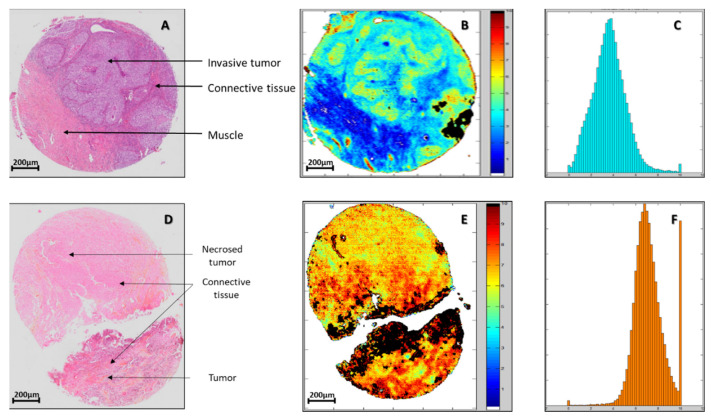
Histology (**A**,**D**); PLS scoring (**B**,**E**) and histogram (**C**,**F**) of test samples from one NR (up) and one R (down) patients. The histograms indicate the number of pixels as function of the R/NR PLS score.

**Figure 3 cancers-14-00021-f003:**
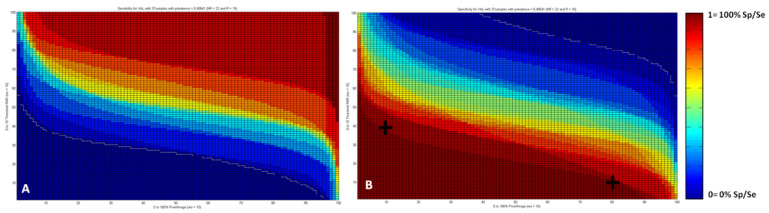
2D map of sensitivity (**A**) and specificity (**B**) for test set according to the values of percentage of pixels (*x* axis) and R/NR PLS score (*y* axis).

**Figure 4 cancers-14-00021-f004:**
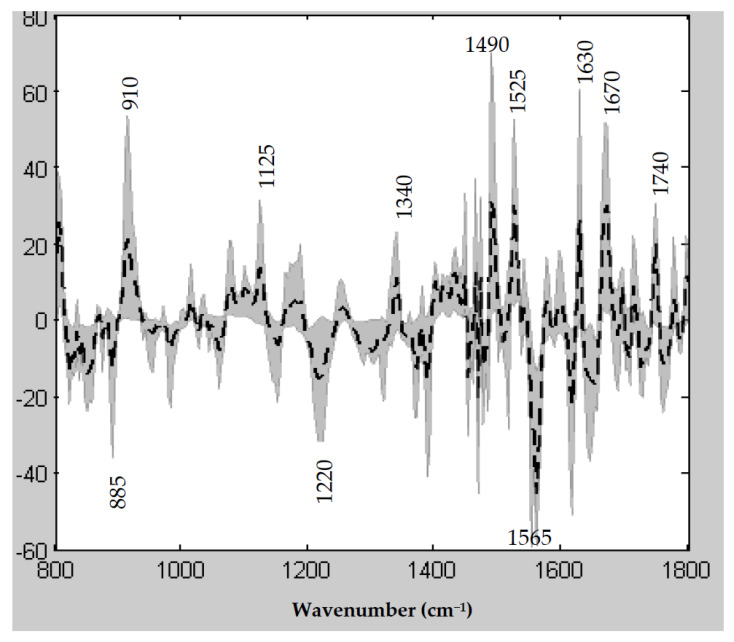
Vibrational infrared features involved in the PLS modeling of the NAC response. The solid line corresponds to the mean of the first twelve PLS latent variables (LV) used in the model and the shaded area represents the minimum to maximum space of variability of these latent variables. The wavenumbers associated with the main spectral features are also indicated.

**Table 1 cancers-14-00021-t001:** Patients characteristics.

Patients Characteristics at Diagnosis	Mean (Lower Quartile–Upper Quartile)
Age	66 (48–78)
Sex	
Male	33 (77%)
Female	10 (23%)
OMS	
0	17 (39%)
1	16 (37%)
2	3 (8%)
Missing data	7 (16%)
Charlson score	3 (2–6)
Smokers	34 (85%)

**Table 2 cancers-14-00021-t002:** Tumor and treatment characteristics.

Treatment and Tumor Characteristics	Number %
Tumor response	
Responders	19 (44%)
Non responders	24 (56%)
Chemotherapy	
MVAC-I	10 (24%)
Gemcitabin cisplatin	24 (57%)
Gemcitabin carboplatin	8 (19%)
Mean number of chemotherapy cycles	4 (3–6)
Toxicities (any grade)	19 (44%)
Time between last chemotherapy and surgery (days)	40 (7–69)
Relapse	
Number	11 (34%)
Distance surgery-relapse (months)	15 (2–37)
Metastatic	10 (91%)
Missing data	11 (26%)

## Data Availability

The datasets used and analyzed during the current study are available from the corresponding author on reasonable request.

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
