# Peer review of "Identification of Neoadjuvant Chemotherapy Response in Muscle-Invasive Bladder Cancer by Fourier-Transform Infrared Micro-Imaging"

_cancers, 2021, doi:10.3390/cancers14010021_

Round 1

Reviewer 1 Report

The authors used FTIR micro-imaging to evaluate the responsiveness of bladder cancer treatment. The method is novel and may hold some potential in preclinical and clinical imaging studies. I have a few questions:

  1. As the authors have noted, the sample size is quite small. Have the authors do any statistical analysis to make sure the differences observed in R and NR groups are statistical significant?
  2. Have the authors considered comparing the FTIR imaging of cancerous tissue with non-cancerous healthy tissues? This might be helpful in filtering out a majority of background signals. 
  3. In terms of FTIR, which are the major frequencies of signals observed? What are the molecules or biomarkers that contribute to the difference of R and NR patients? 
  4. In a practical standard point of view, how will FTIR be used for patient stratification before any treatment occurs? The images presented are FTIR post-treatment, which might be different from those of pre-treatment. Does the imaging work non-invasively, or the tumor tissue needs to be sectioned from the patient? 

Author Response

the responses to the reviewers questions are indicated in the attached Word file

Reviewer 2 Report

In the manuscript “Identification of neo-adjuvant chemotherapy response in muscle-invasive bladder cancer by Fourier-Transform infrared micro-imaging”, the Authors analyzed, by means of mid-infrared imaging, tissue samples of trans-urethral resection of bladder tumor to evaluate the potential of this approach for the prediction of response to neo-adjuvant chemotherapy in muscle invasive bladder cancer.

This work is of great relevance in the field and represents a new potential application of infrared spectroscopy approaches in the clinic.

I have the following comments for the Authors.

1)  In the proposed method, FTIR spectra were analyzed in the 1800-800 cm-1 range and subjected to multivariate analyses. In my opinion, in this type of studies it is mandatory to show representative spectra. Furthermore, it is well known that several multivariate approaches (including PLS-DA) allow to identify the spectral components most relevant for the discrimination, in this case to discriminate between responders and non-responders patients to chemotherapy. This point is particularly important since it allows a better understanding of the involved biological processes thanks to the assignment of the identified spectroscopic peaks.  

2) In general, infrared spectroscopy is a label-free approach that will be ready for clinics after solving several problems (Anal. Chem. 2019, 91, 12117–12128). In this regard, the discussion of works highlighting the potential of IR spectroscopy for different diseases is of great relevance. The Authors should mention more human diseases for which FTIR spectroscopy supported by multivariate analyses have already been proposed. In my opinion, in addition to cancers, it is relevant to add other diseases, such as neurogenerative and other diseases associated with amyloid aggregation (see for instance PNAS 2017, 114 (38) E7929-E7938;  Anal. Chem. 2019, 91, 2894-2900), chronic (see for instance Mol. Biol. Rep. 2016, 43, 1321– 1326) and infectious diseases (see for instance Anal. Chem. 2017,89, 5238– 5245).

The addition of these considerations will further support the promising application of FTIR spectroscopy in clinics.

3) The text should be corrected for typos and other errors. For instance, “,” must be used in place of “;” through the manuscript text.

Author Response

(The authors gave the same response as above.)
